# Climate teleconnections modulate global burned area

Adrián Cardil [1,2,3] ✉, Marcos Rodrigues[4,5], Mario Tapia[2], Renaud Barbero[6], Joaquin Ramírez[2], Cathelijne R. Stoof [7], Carlos Alberto Silva [8], Midhun Mohan[9] & Sergio de-Miguel [1,3] ✉

Climate teleconnections (CT) remotely influence weather conditions in many regions on Earth, entailing changes in primary drivers of fire activity such as vegetation biomass accumulation and moisture. We reveal significant relationships between the main global CTs and burned area that vary across and within continents and biomes according to both synchronous and lagged signals, and marked regional patterns. Overall, CTs modulate 52.9% of global burned area, the Tropical North Atlantic mode being the most relevant CT. Here, we summarized the CT-fire relationships into a set of six global CT domains that are discussed by continent, considering the underlying mechanisms relating weather patterns and vegetation types with burned area across the different world's biomes. Our findings highlight the regional CT-fire relationships worldwide, aiming to further support fire management and policy-making.

Wildfires play an integral role in shaping ecosystems' composition and species evolution at global scale[1], while causing substantial impacts on climate, human health and socioeconomic activities. Previous research has reported a significant decrease in global burned area (BA) due to agricultural expansion and intensification[2]. Yet, the severity and intensity of extreme wildfire events associated with climate change are expected to progressively threaten ecosystems in several biomes, increasing risks to communities worldwide[2,3]. As a consequence, disentangling the drivers governing the global BA is essential to anticipate changes in fire regimes in the forthcoming decades in the face of climate change[4].

Climate teleconnections (CTs; i.e. significant climate responses in a region far from the perturbation source, either concurrent with or time lagged[4,5]) are usually derived from sea surface temperature (SST) or atmospheric pressure variations from seasonal to interdecadal timescales. A number of CTs have been shown to play a prominent role in modulating weather conditions globally[6,7]. These patterns are usually cyclic in nature, and impact ecosystems' primary productivity and fuel moisture, which are key drivers of fire behavior and BA[8,9]. It is also well-known that CTs may influence wildfire activity through immediate to lagged effects by modulating moisture conditions or fuel build-up processes, which are strongly dependent on the environmental conditions of the ecosystems[3,5]. Although the relationships between wildfires and CTs have been studied in several regions across the globe[3–5,10], and the effects of CTs on weather patterns are expected to become more extreme in the future as a result of climate change[4], no global synthesis of synchronous or lagged CT-fire associations in the different biomes worldwide has been addressed so far in the literature.

Here, we explore the complex associations through cross-correlation analysis between the major modes of SSTs and CTs (Fig. 1 and S.1. 2) driving large-scale atmospheric anomalies and global BA from 1982 to 2018 at 0.5° pixel resolution (Fig. 1; FireCCILT11 global BA product[11]), translating them into transboundary CT domains (CTD)

[1]Joint Research Unit CTFC—AGROTECNIO—CERCA, Solsona, Spain. [2]Technosylva Inc, La Jolla, CA, USA. [3]Department of Crop and Forest Sciences, University of Lleida, Lleida, Spain. [4]Department of Geography and Land Management, University of Zaragoza, Zaragoza, Spain. [5]GEOFOREST Research Group, University Institute for Research in Environmental Sciences of Aragon (IUCA), Zaragoza, Spain. [6]INRAE, RECOVER, Aix-Marseille University, Aix-en-Provence 13182, France. [7]Department of Environmental Sciences, Wageningen University, PO box 47, 6700 AA Wageningen, The Netherlands. [8]Forest Biometrics and Remote Sensing Laboratory (Silva Lab), School of Forest, Fisheries, and Geomatics Sciences, University of Florida, PO Box 110410 Gainesville, FL 32611, USA. [9]Department of Geography, University of California-Berkeley, Berkeley, CA 94709, USA. ✉e-mail: acardil@tecnosylva.com; sergio.demiguel@udl.cat

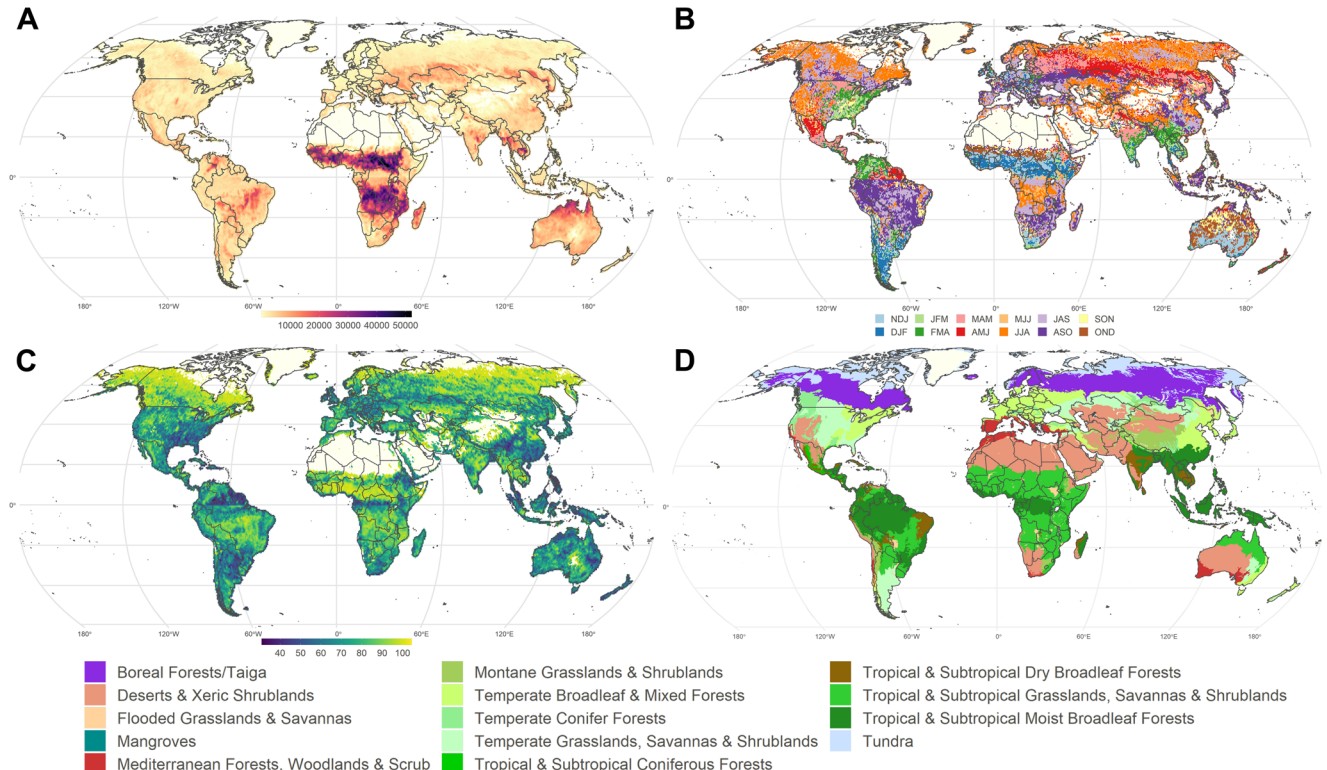

**Fig. 1 | Global burned area (BA), climate teleconnections (CT) and biomes.**
**A** Location of the studied CTs and average annual BA (km²; 0.5° pixel resolution) from 1982 to 2018 during the fire season peak (Atlantic Multidecadal Oscillation (AMO); Arctic Oscillation (AO); East Atlantic (EA); El Niño 3.4 SST index (ENSO); Eastern Pacific (EP); Indian Ocean Dipole (IOD); North Atlantic Oscillation (NAO); Pacific Decadal Oscillation (PDO); Pacific North American (PNA); Southern Annular Mode (SAM); Tropical North Atlantic (TNA); Tropical South Atlantic (TSA); Western Pacific (WP)); (**B**) Fire season peak, defined as the three consecutive months with the highest BA. The capital letters in the legend represent the initial letter of each month; (**C**) Percentage of annual BA during the fire season peak; (**D**) Distribution of the Earth's 14 terrestrial biomes[63]. Fire data: FireCCILT11 global BA product[11]. Projection: Robinson (EPSG:54030).

using a hierarchical clustering approach. These CTD correspond to regions with similar BA responses to CTs at various time lags. Through this study we aim to unravel and clarify the linkages between CTs and BA on different continents and biomes, considering synchronous and lagged signals of CTs not previously analyzed at the global scale (e.g., Arctic and Antarctic modes). This analysis helps to improve our understanding of the role of CTs affecting the natural variability of global BA, allowing for anticipating severe, adverse, and prolonged fire seasons. Our findings are expected to be useful for forest and fire agencies, particularly in emerging regions where long-range danger and risk assessments remain unavailable. The comprehension of these natural system dynamics contributes to the further development of ongoing fire and climate policies and practices aiming to minimize fire-related carbon emissions and negative impacts on societies.

## Results and discussion
Our results revealed statistically significant relationships between CTs and BA worldwide ($P < 0.05$; Figs. 2, 3 and S.1–8), while also identifying regional patterns summarized into a set of six CT domains (CTD; Fig. 3). Most of the CTDs (2, 3, 4, 5 and 6) showed characteristic large-scale CT-fire patterns in specific regions, while CTD 1 had a scattered distribution across global biomes. CTD2 expanded over large areas in southern Australia, South America (Colombia, Equator and Perú), eastern Africa, Ukraine and south-eastern Russia; CTD3 in tropical Africa and America; CTD4 in the African savanna; CTD5 in central and South America and eastern Russia; and CTD6 in southern Africa, northern Australia, Cerrado (Brazil), India and the Asian Boreal taiga. During the peak of the fire season (accounting for 84.3 % of the total annual BA; Fig. 1B, C; see supplementary materials for more details), CTs were responsible

for modulating 52.9% of the interannual variability in global BA ($P < 0.05$; Table 1; Fig. S.9). TNA is the leading CT mode related to BA globally, especially in the northern hemisphere, being associated with 25.7% of the total global BA. Changes in convective activity induced by trade winds and latent heat flux anomalies are thought to modulate the SST variability in the region[12]. TNA dominates variability in terrestrial evaporation and drought in relatively large regions of both hemispheres due to different physical mechanisms explained in the SM[7,13]. In the Southern hemisphere, SAM is the leading CT mode of climate variability associated with changes in the position and strength of the polar jet around Antarctica[14]. This variability drives large fluctuations in weather and climate up from the troposphere to the lower stratosphere[15], which can result in significant linkages with BA, as shown in our analysis. The SAM was associated with 12.3% of the total global BA (Table 1), due to its significant influence across the African savanna, Australia and the South American Savanna (Cerrado; Fig. 2C and S1.8).

Most CTs considered in this analysis featured significant synchronous and lagged signals between CTs and BA. The 0- and 6-month lags were the most important for explaining global BA (Table 1; Figs. 2 and S1.8), as per their influence on weather conditions during the fire season peak (lag 0), and the effect on biomass accumulation and long-term drought leading to dry fuels and flammable landscapes (lag 6). TNA, EA and EP had synchronous and significant correlations with BA, whereas PNA, WP, SAM and NAO were found to predominantly relate to BA through lagged effects. Our study confirms and further expands our knowledge on the prominent role of CTs in modulating global BA through important remote climate feedbacks operating at different spatio-temporal scales, either concurrent or asynchronous, and entailing regional to global impacts. These connections directly

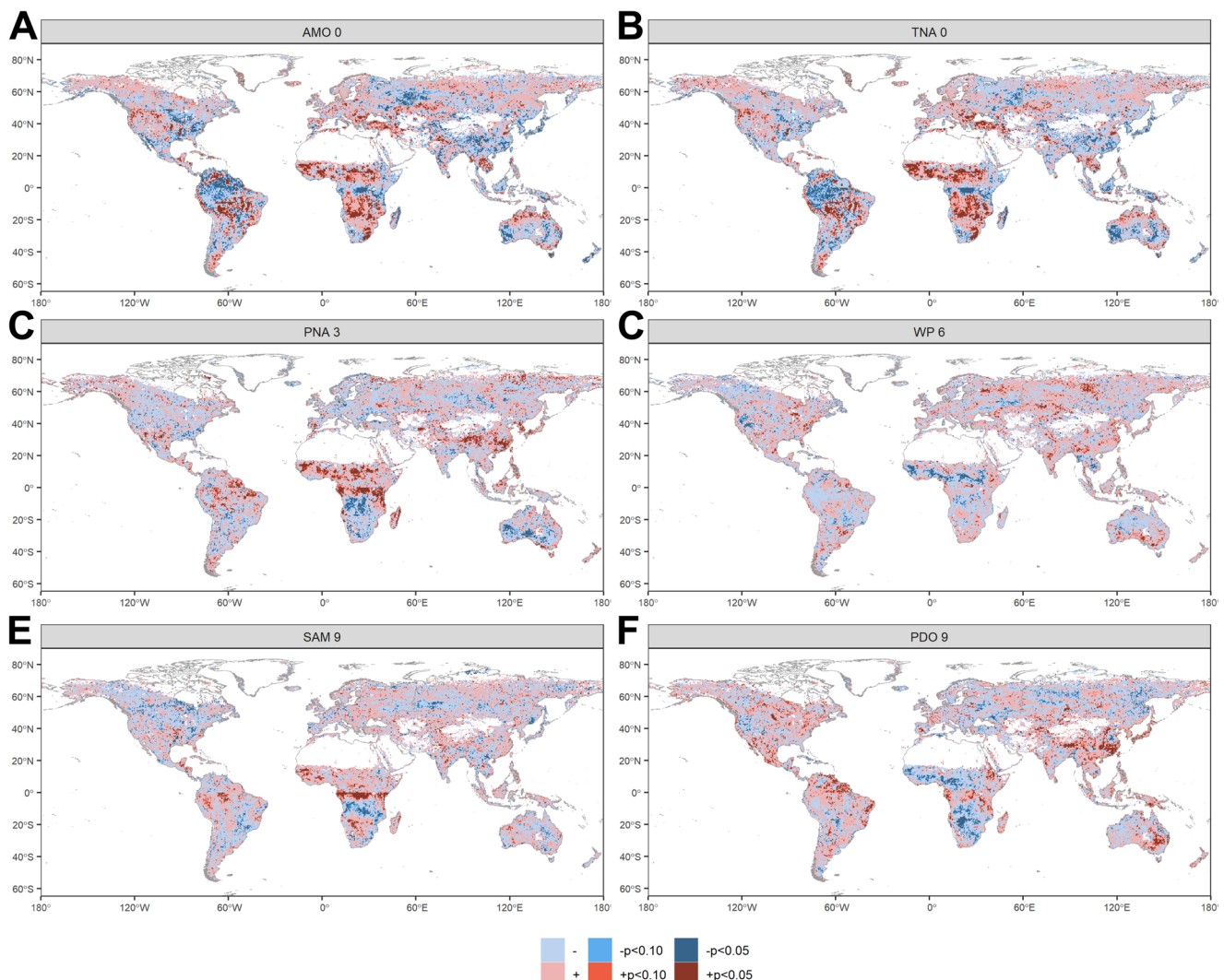

**Fig. 2 | Synchronous (0-month lag) and lagged (3, 6 and 9 months) Pearson R correlation between climate teleconnections (CT) and burned area (BA) during the fire season peak from 1982 to 2018.** The maps at 0.5°x 0.5° pixel resolution represent the correlations at $P < 0.05$ ($|R| > = 0.34$) and $P < 0.10$ ($|R| > = 0.29$) between BA and CTs. The six most important CTs in terms of modulated global BA are presented, namely Atlantic Multidecadal Oscillation (AMO (lag 0)), Tropical North Atlantic (TNA (lag 0)), Pacific North American (PNA (lag 3)), Western Pacific (WP (lag 6)), Pacific Decadal Oscillation (PDO (lag 9)) and Southern Annular Mode (SAM (lag 9)). See the supplementary materials for other CT modes and lags.

influence biomass accumulation and moisture content associated with changing weather conditions.

The spatial extent of influence of each CT varied from sub-continental (WP, EP, EA, AO, IOD) to global scale (AMO, TNA, TSA, NAO, PNA, PDO, ENSO, SAM). Altogether, more than half of the global interannual BA (52.9%) can be explained by CTs. This is slightly higher than previously thought (48.0%) as prior studies did not include the essential effect of SAM[8], affecting the major BA hotspots in the southern hemisphere due to changes in the strength and position of the polar jet around Antarctica[14]. Our analysis also shows that global BA is largely explained by TNA and AMO, rather than by ENSO, as previously thought[8]. However, although our analysis did not find any significant correlation between TNA and ENSO (Fig. S1), some studies suggest that the low-frequency component of ENSO in the Pacific is associated with TNA anomalies through inter-basin connections[16].

The six fire-CT domains identified in this paper allow for better comprehension of regional to global fire activity, which is useful to anticipate BA, adverse fire seasons and forecast which areas of the world are likely to burn simultaneously in the future[8]. As such, the CT associations with BA highlighted in our analysis

provide valuable information for land managers and emergency services worldwide, particularly in regions where long-range fire risk forecasts are not readily available. The percentage of BA modulated by CTs is high, given that fire is not just controlled by global climate but also regionally by land management, fire suppression capabilities and practices.

There is increasing evidence that some CTs are non-stationary, strengthening or weakening their signal on multi-decadal timescales; behavior which may further influence climate and BA[6]. The source of these low-frequency changes is still unclear but may arise due to internal variability of the climate system or anthropogenic climate change[17]. In fact, the trends of some CT indices is a well-established phenomenon and has already been linked to climate change[17]. For instance, previous research identified an intensification of TNA variability under climate warming. This enhanced variability may strengthen the ENSO-forced PNA pattern and tropospheric temperature anomalies due to an eastward shift of ENSO-induced equatorial Pacific convection and of enhanced ENSO variability, which may reinforce the associated wind and moist convection anomalies[18]. Also, a robust positive trend in the SAM is projected for the end of the 21st century under different climate change scenarios due to its response

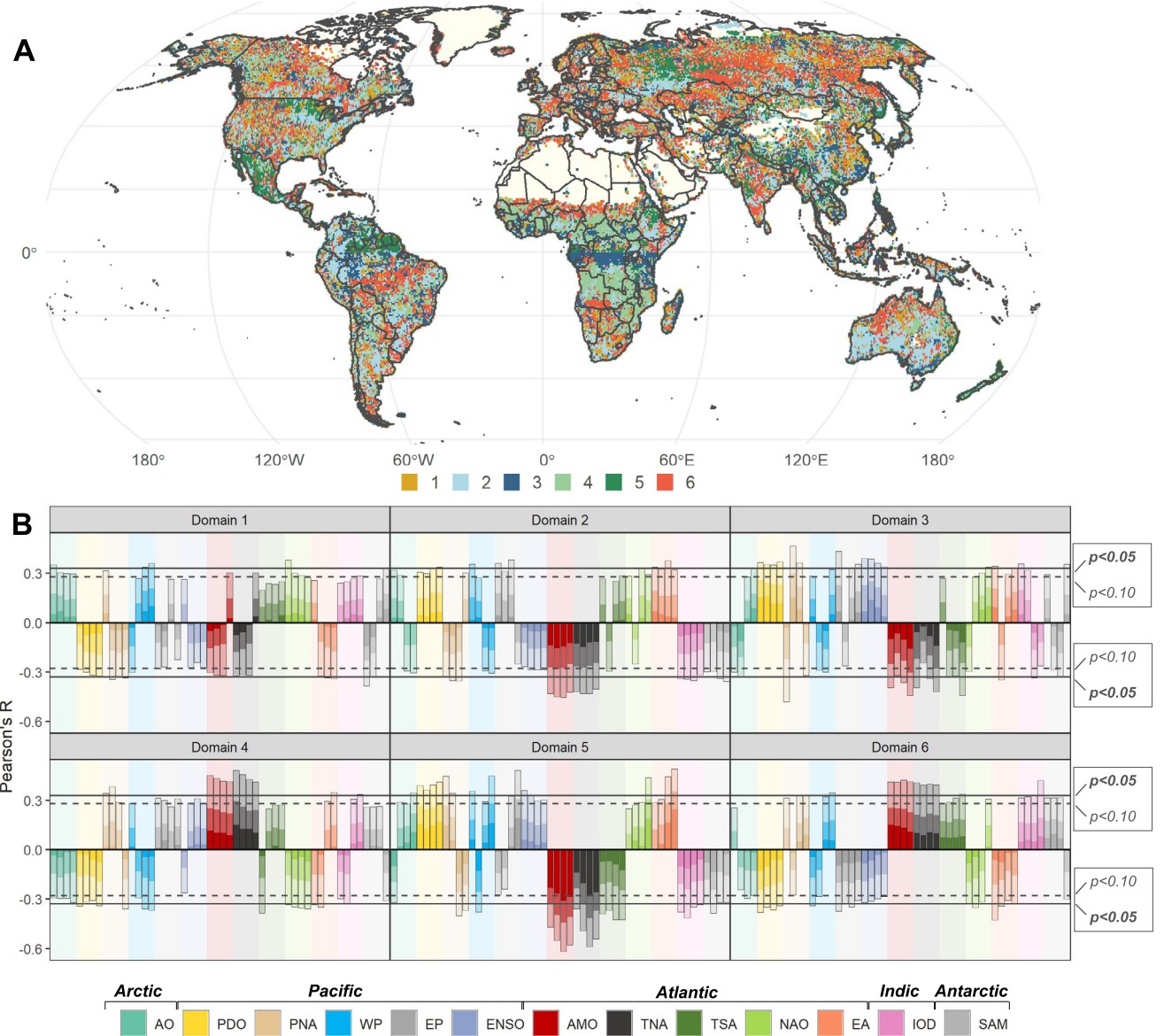

**Fig. 3 | Climate teleconnection domains (CTD) characterizing CT-fire associations worldwide.** These domains were derived by applying a hierarchical clustering approach to the correlation coefficients obtained for the different combinations of CTs and lag windows at pixel level. **A** Spatial distribution of the six CTD. **B** Characterization of the CTD based on the relationship between each CT and burned area at a given time lag. Colored bars indicate the corresponding CT; color brightness relates (from darker to lighter) to the median, 75th and 95th percentile of Pearson's R. Correlative bars with the same color are ordered from lag 0 to 9. Dashed lines mark the significance threshold ($P < 0.05$ ($|R| > 0.34$) and $P < 0.10$ ($|R| > 0.29$)). Atlantic Multidecadal Oscillation (AMO); Arctic Oscillation (AO); East Atlantic (EA); El Niño 3.4 SST index (ENSO); Eastern Pacific (EP); Indian Ocean Dipole (IOD); North Atlantic Oscillation (NAO); Pacific Decadal Oscillation (PDO); Pacific North American (PNA); Southern Annular Mode (SAM); Tropical North Atlantic (TNA); Tropical South Atlantic (TSA); Western Pacific (WP).

to stratospheric ozone depletion over Antarctica[14], which could result in rainfall decreases in the midlatitudes and increases in the high latitudes of the southern hemisphere, leading to potential and dramatic changes in BA at sub-continental scales[15]. Also, CTs may show asymmetric behavior in magnitude, duration or seasonal timing in relation to their positive and negative phases, resulting in climate responses that may be non-linear or sensitive only to a specific CT phase[19]. In light of the uncertain global changes in the forthcoming decades, it is wise to recognize that the observed relationships reported in this study may change in strength and nature in the future.

Below, we highlight the main findings by continent and biome to facilitate interpretation considering the underlying weather patterns behind the CT-fire associations at subcontinental scales. Extended insights into global CT-fire relationships and potential mechanisms explaining these patterns are shown in the Supplementary Materials due to space constraints.

## Africa

Africa accounts for about 70% of BA[11] and 50% of fire carbon emissions worldwide[20]. Correlations between weather patterns and some CTs have been recently investigated[13], but few studies have addressed the linkage between CTs and wildfires in Africa[4]. Most fires occurred in areas occupied by savannas (CTD 4). However, other biomes such as the tropical moist forest of the Congo Basin (CTD 3), one of the most biodiverse areas on Earth, also deserve special attention given the increasing anthropogenic pressure it is experiencing.

Savanna fires responded mostly to +TNA (lag 0 and 3), +AMO, +SAM (lag 9), +EA (lag 9) and +ENSO in the northern hemisphere (CTD

**Table 1 | Climate teleconnections (CTs) modulate 52.9% of the global burned area (BA)**

|  | Lag 0 | Lag 3 | Lag 6 | Lag 9 | Any lag |
|---|---|---|---|---|---|
| TNA | 18 | 13.7 | 10.7 | 10.4 | 25.7 |
| PNA | 5.6 | 12.5 | 4 | 6 | 24.3 |
| AMO | 15.9 | 13.3 | 12.1 | 12.3 | 22.3 |
| WP | 2.9 | 2.6 | 7.1 | 6 | 16.6 |
| NAO | 4.8 | 3.5 | 5.7 | 4.2 | 16.1 |
| EA | 5.6 | 4.6 | 4 | 5.3 | 16 |
| EP | 5.8 | 5.3 | 2.7 | 3.1 | 15.4 |
| PDO | 4.9 | 4.1 | 4.4 | 7.5 | 12.5 |
| SAM | 2.3 | 2.2 | 2 | 6.6 | 12.3 |
| AO | 3.2 | 2.4 | 3 | 2 | 10 |
| IOD | 2.8 | 2.7 | 2.8 | 2.7 | 9.1 |
| ENSO | 3.2 | 3.8 | 3.5 | 2.7 | 8.1 |
| TSA | 2.2 | 2.9 | 2.4 | 3.1 | 8 |
| Global |  |  |  |  | 52.9 |

Percentage of annual BA modulated by the corresponding CT and studied lag (synchronous and lagged signals) as represented by those pixels with statistically significant correlation ($P < 0.05$) between CT indices and BA during the fire season peak.

*AMO* Atlantic Multidecadal Oscillation; *AO* Arctic Oscillation; *EA* East Atlantic; *ENSO* El Niño 3.4 SST index; *EP* Eastern Pacific; *IOD* Indian Ocean Dipole; *NAO* North Atlantic Oscillation; *PDO* Pacific Decadal Oscillation; *PNA* Pacific North American; *SAM* Southern Annular Mode; *TNA* Tropical North Atlantic; *TSA* Tropical South Atlantic; *WP* Western Pacific.

4); +AMO, +TNA (lag 0, 6 and 9), -SAM (lag 9), +PNA (lag 0), −PNA (lag 3) and -NAO (lag 6 and 9) in the southern hemisphere (CTD 4); both +IOD and +ENSO conditions in East Africa (CTD 2), related to above-average rainfall, were negatively correlated with BA[21], specifically in Kenyan and Ethiopian subtropical grass-savanna biome (lag 0, 3 and 6). Our analysis revealed significant lagged CT-fire relationships, in accordance with fire dynamics in this biome, driven by a rapid fuel build-up during the wet season and by a warm season capable of drying fuels out. +TNA (lag 0 and 3) and +AMO were correlated with increased BA likely due to changes in atmospheric circulation driving lower precipitation[22]. The +SAM (lag 9) was strongly correlated with increased BA in the CTD 4 (northern savanna; lag 9) and decreased BA in CTD 4 (southern savannas) likely due to altered precipitation and vegetation moisture[15]. Also, in the CTD 4 in the southern hemisphere, +NAO (lag 6 and 9) was associated with reduced BA, which is consistent with the fact that it promotes wetter conditions prior to the fire season peak[23]. +PNA (lag 3) was associated with decreasing BA as it is potentially linked to above average convective precipitation over central Africa during the monsoon season.

We found that BA in the tropical moist forest biome (CTD 3) was mainly associated with -TNA (lag 0) and PNA (lag 0, 3, and 9), which promoted wetter conditions while BA diminished, and +SAM (lag 9) which promoted drought conditions thereby possibly boosting BA[24]. In southern Africa (CTD 1, 2 and 6; Fig. 1D), we found significant effects of TNA, AMO, ENSO (lag 3, 6 and 9), PNA (lag 0), NAO (lag 9), AO (lag 9), EA (lag 0 and 9), EP (lag 0), IOD (lag 0 and 6), and SAM (lag 9) on BA. The +SAM (lag 9) promoted BA due to reduced rainfall in the austral summer in Angola (CTD 6). PNA (lag 3) was positively correlated with BA because of increased precipitation promoting biomass before the fire season. NAO and AO influenced precipitation with a marked dipole in CTD 2 and 6. In CTD 6, precipitation during austral winter (lag 6 and 9) increases with +NAO, whereas the opposite occurs in CTD 2, influencing BA accordingly[23].

## Americas

Wildfires in the Americas affected a diverse range of biomes each displaying unique fire regimes and peak season timing. Most of the fires occurred in the tropical savannas (Colombian Llanos, CTD 3, and

Cerrado CTD 6), where teleconnections such as +PDO and +ENSO, linked to anomalously warm and dry conditions[25], concurred in correlation with increased BA, while +IOD and +SAM, which are known to foster precipitation, were both associated with reduced BA[25]. +TNA and +AMO drive increases in BA in CTD 6 probably due to significant increases in terrestrial evaporation during and before the fire season peak[7].

Amazon fires (CTD 2) threaten one of the most megadiverse regions in the world. CTs associated with increases in BA included the +AMO, +TNA, +NAO and the +SAM, particularly at 9-month lag[25]. Drought is mediated by +TNA and +AMO associated with a sustained northward position of the ITCZ[26]. +PNA is accompanied by positive precipitation anomalies, diminishing fire activity, in the austral summer[27]. In the southern end of the continent, Patagonia and the Pampas (CTD 2 and 6), within the temperate grassland and shrubland biome, +SAM is known to boost wildfire activity by promoting significantly warmer and drier conditions during the austral summer[25,28].

Wildfires within the North American taiga biome are a dominant disturbance shaping landscape diversity. The positive phases of PDO, ENSO, and IOD correlate with anomalously warmer and drier conditions across all of the Pacific Northwest and Canada, and are likely responsible for the observed increase in BA at the 9-month lag[29,30]. The +SAM is known to drive cooler temperatures across northern latitudes in North America, which is reflected in the decrease in BA at the 9-month lag[27]. Also, EA, AO, NAO, EP and WP correlate with BA at different time lags due to changes in terrestrial evapotranspiration as found by previous research[7]. The southeastern United States hosts a number of primarily temperate ecoregions. The positive phases of NAO, EA and ENSO were all positively correlated with BA through producing cooler and wetter conditions as identified by previous research[31]. CT-fire relationships in western United States were generally weak, mainly related to +AMO, +PDO and +ENSO, which may induce drought conditions and increased BA. BA in Southern Mexico and Central America (CTD 5) is correlated with the positive phases of AMO, TNA, ENSO and PDO, all of them associated with drought conditions[32].

## Asia

Asia's major BA hotspots are located in the temperate steppe (central Asia), the tropical and subtropical dry forests of southeastern Asia (India, Thailand) and the temperate broadleaf forests of China. BA in southern Asia (CTD 5 and 6), mainly correlated with -ENSO (lags 0, 3 and 6), -PDO, -IOD (lag 3), AMO, TNA, +SAM (lags 0 and 3), TSA (lags 6 and 9), +EA (lag 6), -EP (lag 3) and +WP (lag 6 and 9), with a fire season extending from January to May varying latitudinally across the continent (Fig. 1). BA across mainland SE Asia correlated with +ENSO (lag 0, 3) +PDO, +IOD (lag 3, 6 and 9), +EP (lag 9), +TNA (lag 0, 3), +AMO and +EA (lag 3). +PDO influences the East Asian Summer Monsoon, modulating drought across the region[33], +ENSO amplifies frequency, duration and intensity of heat waves and drought in much of Indochina, therefore likely contributing to increases in BA[34]. +IOD restricts rainfall in June-August, particularly in Cambodia and central Laos, where we witness the greatest BA increases. Wildfire activity in the Maritime region of SE Asia, often related to peatland fires, was predominantly associated with drought mediated by the positive phases of ENSO (lag 0, 3 and 6), TNA (lag 0), IOD (lag 0), PNA (lag 6), WP (lag 9).

Eastern Asia (China) holds the world's largest subtropical forests, and falls mostly within the CTDs 1, 2, 3 and 6, with a fire season peak timing varying latitudinally (Fig. 1). We found significant effects of +ENSO on BA (lag 6 and 9), probably amplified by +PDO[35]. Reduction in BA was found under +AMO and +TNA (lag 0), which have been reported to enhance precipitation over the region[36]. Also, +EA (lag 0), +EP (lag 0) and +WP (lag 9) correlated with increased BA, all these

driving weather conditions of limited precipitation, high temperature and dry fuels during fire seasons[37,38].

The Asian Boreal taiga is a vast landscape dominated by coniferous forests. The western taiga across the Ural Mountains (CTD 6) experienced heightened BA during the positive phase of AO (lag 0, 3 and 9), NAO (lag 3 and 9), EA (lag 9), WP (lag 6 and 9) and SAM (lag 6). Reductions in BA correlated with the positive phases of IOD (lag 0, 3 and 6) and AMO. Eastern Siberia (CTD 5) appeared to be susceptible to increases in BA during +NAO, +AO, +PNA (lag 3, 6) and +IOD (lag 0) events. We also observed similar impacts of AO and NAO (both highly correlated, partly due to shared storm tracks) between the northeastern Atlantic and Arctic regions during the wintertime[39]. NAO was found to be positively correlated with BA at 9-month time lag, which may indicate fuel build-up connecting to fire activity within the Siberian taiga.

The Eurasian Steppe (or Great Steppe) is among the largest grasslands in the world, extending from Hungary to China. Around 90% of BA in Central Asia was located within Kazakhstan (CTD 2 and 6)[11]. AMO, TNA, PNA and EA were the most prominent teleconnections related to wildfire activity in this region. Spring EA is associated with decreased summertime precipitation and reduced aboveground net primary productivity[40]. It has also been observed that a significantly higher sea level pressure and geopotential height in the summer following a warm winter AMO could lead to adiabatic warming and elevated risk of wildfires.

## Europe

Approximately, half a million hectares are burned across Europe every year. More than 85% of the BA is located in southern Europe, a Mediterranean fire-prone area[11]. CT-fire relationships in Europe were weaker compared to other continents, because a considerable part of the continent is under relatively low-fire prone climate regimes in a fragmented landscape, which contributes to fading these relationships.

NAO is among the most notable CTs with significant influence within Europe[5]. +NAO is associated with northward shifts in the location and intensity of the North Atlantic jet stream responsible for driving moisture transport across Europe. +NAO correlated with reduced BA in temperate continental and Scandinavian Europe (lag 3, Fig. S4), likely owing to increased westerly winds, which bring moist air into Europe producing mild winters accompanied with frequent rain[6] prior to the fire season peak. +EA, often in its connection with NAO, correlated with increasing BA across central and western Europe (lag 9) likely due to warm temperature and low precipitation anomalies in late spring[41]. In contrast, +EA was associated with decreased BA in eastern Europe. Fires in the Mediterranean and temperate regions were mostly related to +PDO, +IOD and +ENSO, which are linked to anomalies in precipitation, promoting vegetation build-up before the fire season peak[5,42]. TSA positively correlated with BA in the Spanish Mediterranean coast (lag 9), whereas +AMO and +TNA (lag 3, 6 and 9) were predominantly associated with increased BA in Greece and Turkey.

## Oceania

The 2019–2020 extraordinary Australian wildfires showed how climatic events can cause unprecedented large-scale impacts through the combination of sustained record-breaking high temperatures with low precipitation[10]. The equatorial Pacific SST variability is the main driver of observed precipitation patterns although the Indic and Atlantic Oceans also influence Australian precipitation and temperature by altering the tropical Pacific climate through the trans-basin variability mechanism[43,44]. Two CTDs (2, and 6; Fig. 3) encompassed the main CT-fire relationships in this continent. In northern Australia (CTD 6), BA is enhanced by +AMO, +TNA (lag and 9), +IOD (lag 0, 3 and 6) and -PDO. In the CTD 2, in addition to the effects of the aforementioned CTs, we found an effect of +ENSO in southeastern Australia that brings

increased surface air temperature and drought conditions[45], potentially coupled with +IOD[46]. +IOD brings easterly wind anomalies across the Indian Ocean that diminish rainfall over Australia[46]. Precipitation is enhanced over northern Australia by +PDO in contrast to the eastern and western ends where it is weakened[40]. TSA positively correlated with BA in southeastern Australia probably due to increased surface air temperature during the Austral summer. As expected, the effects of TNA and AMO in this region attained opposite signs. Also, +EP (lag 3), +WP (lag 6), PNA (lag 3 and 6), +EA (lag 3 and 6) and +NAO (lag 6) appear to relate to increases in BA in the CTD 2 as found by Shi and Touge (2022)[47]. +SAM (lag 6) was found to be negatively correlated with BA, likely due to concurring cool and wet conditions over southeastern Australia[28].

## Methods

### Fire data

We analyzed global burned area (BA) for the period 1982–2018 (Fig. 1A) from the FireCCILT11[11], developed within the European Space Agency's (ESA) Climate Change Initiative (CCI) program under the Fire Disturbance project (https://climate.esa.int/en/projects/fire/, last access: 3 June 2021). The product was assembled using spectral information from the AVHRR sensor (Advanced Very High-Resolution Radiometer) and Land Long Term Data Record (LTDR) version 5 dataset produced by NASA (National Aeronautics and Space Administration). The FireCCILT11 includes monthly global BA data estimates at 0.05° cell resolution. Data was resampled at 0.5° resolution based on the monthly scale to facilitate data handling and processing while enhancing pattern recognition. There are several existing global BA datasets that could have been used in this study[47]. However, we selected the FireCCILT11 global BA product as the most suitable dataset because the time series is long and it performs better than other global BA products in terms of small wildfire detection capacity[47].

### Climate teleconnections data

Using a 3-month moving average, we analyzed the main climate teleconnections (CTs) modulating weather patterns at global scale known to influence wildfire activity from regional to continental scales[3,5,10,28,48], namely:

The Atlantic Multidecadal Oscillation (AMO) index refers to Sea Surface Temperature (SST) variation of the North Atlantic Ocean (0-70°N) on the timescale of several decades[49].

Arctic Oscillation (AO) teleconnection or Northern Annular Mode (NAM) is highly related to the NAO and is derived as a function of winter sea level pressure (geopotential heights) and surface air temperature fluctuations over mid to high latitudes (>20°N) in the Northern Hemisphere[50].

The East Atlantic (EA) teleconnection pattern index consists of a north-south dipole of sea level pressure anomaly centers spanning the North Atlantic from east to west and is often drawn comparisons with the NAO due to their similar structure[51].

The El Niño 3.4 SST index, also commonly referred to as ENSO, is derived from HadISST1 and is calculated as a function of area averaged sea surface temperature anomalies (SST) from 5°S–5°N and 170°–120°W[52].

The Eastern Pacific (EP) teleconnection pattern is a convective dipole and it has two primary sea level pressure centers in the North Pacific and two secondary centers in North Asia and subtropical Pacific[53].

Indian Ocean Dipole (IOD) is defined as the difference in SST between the tropical west Indian Ocean (50°E-70°E, 10°S-10°N) and the tropical southeastern Indian Ocean (90°E –110°E, 10°S-equator)[54].

The North Atlantic Oscillation (NAO) index is based on the differences in atmospheric pressure at sea level between Gibraltar and Reykjavik and is distinguished for its strong influence on climate variability in the Northern Hemisphere[55].

The Pacific Decadal Oscillation (PDO) index phases are defined as cool and warm regimes based on monthly SST anomalies in the North Pacific Ocean, poleward of 20°N, and are known for driving interdecadal climate variability across the Pacific[56].

The Pacific North American teleconnection pattern (PNA) is among the more noteworthy modes of low-frequency variability in the Northern Hemisphere extratropics. The PNA is linked with strong fluctuations in the magnitude and location of the East Asian jet stream[51]. The PNA index is obtained by projecting the PNA loading pattern to the daily anomaly 500 millibar height field over 0-90°N. Southern Annular Mode (SAM), otherwise known as the Antarctic Oscillation (AAO), is the dominant mode of atmospheric variability in the Southern Hemisphere and is calculated as the difference in zonal mean sea level pressure (SLP) between 40°S and 65°S[57,58].

The Tropical North Atlantic (TNA) index is defined as monthly SST anomalies in the Atlantic Ocean from 5.5°N to 23.5°N and 15°W to 57.5°W. This SST anomaly affects the atmospheric pressure gradient and translates into trade wind anomalies near the equator[59].

The Tropical South Atlantic (TSA) index is defined as monthly SST anomalies in the Atlantic Ocean from Equator to 20°S and 10°E to 30°W[59].

The Western Pacific (WP) teleconnection pattern is defined as a geopotential height anomaly, characterized by jetstream movements. Its dipole is distributed from the Kamchatka peninsula to southeastern Asia[60].

The CT indices were retrieved from the National Centers for Environmental Information and National Weather Service (NOAA).

## Climate teleconnections associations
We analyzed the relationships among the CTs during the study period (1982–2018) using Principal Component Analysis (PCA) and Pearsons's correlation coefficients. We found that most of the CT indexes were not significantly correlated except for both ENSO/PDO ($r = 0.42$), AMO/TNA/TSA ($r = 0.83$) and AO/NAO ($r = 0.60$), which were positively and significantly ($P < 0.05$) correlated. PCA biplots further support these links with TNA/AMO, ENSO/PDO and AO/NAO being clearly aligned in the PC1–PC2 space (Fig. 4).

## Statistical analysis
Several analyses were conducted to test for significant relationships between CTs and BA. We first identified the fire season peak at pixel level across the globe (0.5° x 0.5° resolution) to focus on the main fire season. Then we conducted a cross-correlation analysis calculating the Spearman's *rho* and Pearson's R correlation coefficients between annual time series of BA in fire season peak and CTs. Finally, we synthesized the observed associations into climate teleconnection domains (CTD) that represent characteristic large-scale CT-fire patterns from local to subcontinental scales. All statistical analyses and tests were conducted using the R software[61] with a confidence level of 95% ($P < 0.05$).

The timing of the main fire season varies spatially across the globe. To properly assess the influence of CTs on BA, we therefore first analyzed the spatial timing of the main fire season, defined as the 3 consecutive months accumulating the highest average BA (Fig. 1B). We calculated this fire season peak at pixel level to subsequently analyze CT-fire associations, including synchronous or lagged relationships due to fuel built-up processes, and weather conditions influencing fire behavior. Finally, we calculated the percentage of annual burned area during the fire season peak to highlight the relevance of the analysis in each region (Fig. 1C). The fire season peak serves as a temporal baseline to implement cross-correlation analysis, corresponding to lag 0 or synchronous association in the cross-correlation analysis.

To analyze the linkages between CTs and BA, both the Spearman's *rho* and the Pearson's R correlation coefficients were calculated between annual time series of BA during the fire season peak and CTs at pixel level (0.5° x 0.5° resolution). In addition to synchronous correlation (i.e., lag = 0 or synchronous to the fire season peak), we investigated asynchronous effects of the CTs' signal on BA by exploring a lag of 3, 6 and 9 months before the fire season peak time. The spatial pattern of correlations was summarized in maps depicting the direction (either positive or negative) and the significance ($p < 0.05$ and $p < 0.1$) to better display the CT-fire association. Time series were detrended before correlation analysis by implementing a Seasonal and Trend Decomposition (STL), i.e., subtracting the "trend" component from the original time series[48]. The STL procedure was applied to both CT and BA monthly time series before computing the lagged averages. The procedure consists of splitting the original series into its 'trend', 'seasonal' and 'random' components, retaining only the latter two, i.e., removing the tendency in the time series. The procedure was conducted at pixel level in the case of BA data, and globally for each CT pattern. Once the STL was performed, the detrended time series were

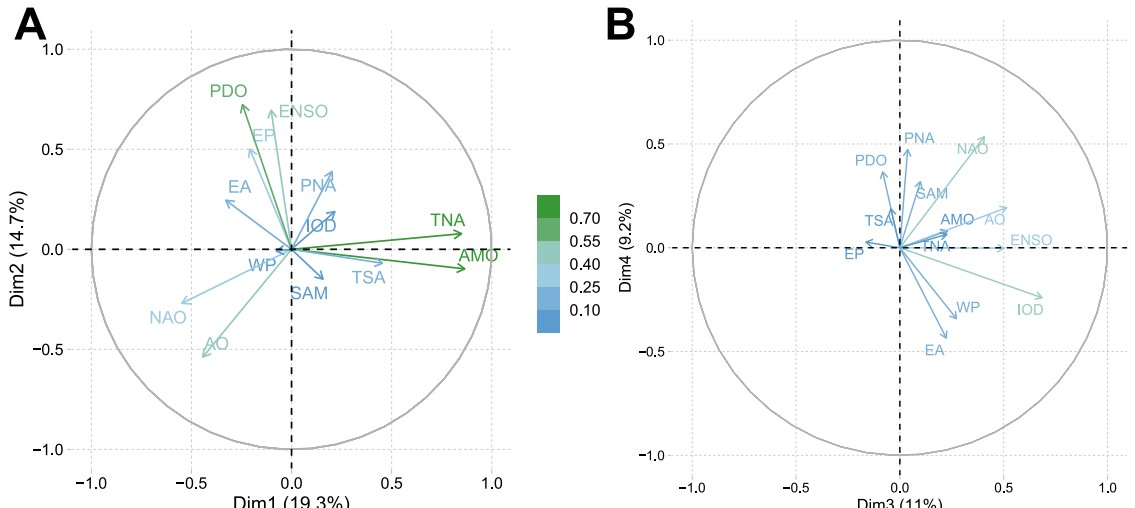

**Fig. 4 | Relationships among climate teleconnections (CT) from 1982 to 2018 using Principal Component Analysis (PCA). A** First versus second PCA component (34.0% of variance); (**B**) third versus fourth PCA component (20.2% of variance). Color indicates the importance of the component for each variable as the square cosine.

aggregated as a 3-month average and correlation was calculated between seasonal BA during the fire season peak and the corresponding 3-month lagged interval (3, 6 and 9 months before the fire season peak) of CT indexes. BA was log-transformed to achieve a normal distribution, as required for Pearson's R linear correlation analysis.

Finally, in order to better understand the relationship between climatic forcing and BA at multiple time scales, we summarized individual CT-fire associations into a set of CTDs. These CTD were derived by applying cluster analysis to the Pearson's correlation coefficients obtained for the different combinations of CTs and lag windows at pixel level. We synthesized the association with the four analyzed time lags, grouping pixels into clusters depicting similar linkages. A hierarchical clustering approach was implemented in combination with *ward.D2* acting as agglomeration criterion and *Canberra Distance* as a measure of dissimilarity to minimize intergroup variance. The number of clusters (i.e. climate domains) was optimized in accordance with all the criteria available in the *nbClust* R package[62]. The observed distribution of CT-fire correlations was used to characterize the resulting climate domains, highlighting the most prominent driving patterns in terms of influence and direction of the relationship at a given time lag. To do so, we displayed the pixel-based correlation distribution at each CTD and lag, highlighting their significance level. We mapped the spatial distribution of the resulting CTDs.

### Reporting summary
Further information on research design is available in the Nature Portfolio Reporting Summary linked to this article.

## Data availability
The global burned area data from the FireCCILT11[11], developed within the European Space Agency's (ESA) Climate Change Initiative (CCI) program under the Fire Disturbance project is publicly available (https://climate.esa.int/en/projects/fire/, last access: 4 December 2022). The climate teleconnection indices were retrieved from the National Centers for Environmental Information and National Weather Service (NOAA; https://www.cpc.ncep.noaa.gov/, last access: 4 December 2022).

## Code availability
The processing codes used in this study are available from the corresponding authors upon request.

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

## Acknowledgements

We thank Lorea Garcia for her insights and useful suggestions in the interpretation of CT-fire relationships during the review process of the manuscript. This project received funding from the Spanish Ministry of Science and Innovation, project FIREPATHS (PID2020-116556RA-I00) (authors receiving funding: A.C. and M.R.) and the European Union's Horizon 2020 research and innovation programme MSCA-ITN-2019—Innovative Training Networks under grant agreement No. 860787 (PyroLife) (authors receiving funding: A.C., M.T. and C.S.), and the European Horizon 2020 research and innovation programme under grant agreement No. 101037419 (FIRE-RES) (authors receiving funding: A.C., J.R., C.S. and S.d.M.).

## Author contributions

A.C., M.R. and S.d.M. designed and drafted the manuscript and conducted the analyses. A.C., M.R., M.T., R.B., J.R., C.S., C.A.S., M.M. and S.dM commented and revised the manuscript.

## Competing interests

The authors declare no competing interests.
