## [Peer Review File · Nature Communications]

Reviewer comments, first round -

Reviewer #1 (Remarks to the Author):

Large-scale climate anomalies play key roles in modulating wildfire burnt area globally. This study aims provide a global picture of the influences of CT on burnt area. This study suggested a stronger impact of CT on fire than previous studies. This work is important. My comments and suggestons are shown below.

Many CTs are correlated, such as the enso and sam. How to distinguish their auto-correlations when you evaluate their contributions to the global BA? In addition, a regional BA can be influenced by different CT in different time. It is also needed to consider this.

Some of the CTs you mentioned are widely recognized as CT but some are not. Some patterns have both regional impacts and teleconnections. Please specify how do you distinguish the impacts of regional impacts and teleconnections. For example, the enso and pdo you selected are highly correlated.

Please explain the selection of these ct patterns. For example, why do you selected pdo but not amo? Why do you consider SAM but not NAM? There are also other ct patterns.

Line 294, they are indeed negatively correlated. How to disentangle their impacts?

Line 303-304, it is difficult to understand that the human induced ignition can amplify the relationships between ct and climate. The human caused ignitions can increase the BA but not their linkages with climate.

Please add more details in the figure captions.

It is suggested to add more details on the methods and data. Different fire datasets have different quality of the BA.

Reviewer #2 (Remarks to the Author):

Summary

The manuscript presents a global relationship between weather conditions and fire activity, which vary across continents and biomes. The study find that the 65% of global burned area (peak of the fire season) is modulated by seven global climate connections. The global analysis focuses on regional patterns, appropriately using the burned area database. The method is based on consistent statistical metrics, offering results $p < 0.05$ which is statistically significant. In addition, the study has been widely discussed with an appropriate literature. Therefore, the manuscript presents a novel and relevant study/results that are very interesting for the science community. It helps us improve our understanding of climate and its relationship to fires.

Major comments

The main comment is about the burned area dataset used in the analysis. In the study, FireCCILT10 is mentioned as the product used, but the bibliographic reference corresponds to the FireCCILT11 product. Please, what database has been used? FireCCILT10 is a deprecated Beta version with serious limitations that should no longer be used. On the contrary, FireCCILT11 is a consistent version that improves and fixes the problems of the previous Beta version. The FireCCILT11 product is freely available and suitable for this study. If FireCCILT11 is used, as expected, please changed the information in the manuscript and in the Supplementary Materials. As a suggestion, FireCCILT11 covers the period from 1982 to 2018 (excluding 1994). Why did the study start in 1995? and not in 1982? More years should be interesting for this manuscript and

could offer strong patterns. Despite that, the current time series (1995-2018) shows very interesting results.

Regarding the product nomenclature, the correct way to reference the product, recommended by Fire_cci project (European Space Agency), is FireCCILT11 and not FireCCILT11v1.1. The number 11 corresponds to the version number.

In Supplementary materials, line 17-18: The current website of the Fire_cci (ESA) project is: <https://climate.esa.int/en/projects/fire/>. Please change it.

In Supplementary materials, line 19-20: It is necessary to clarify that AVHRR is a sensor and LTDR is the dataset. Therefore, the LTDR dataset is based on images of the AVHRR sensors.

In Supplementary materials, line 21-22: Clarify that FireCCILT11 includes monthly global BA data estimates at 0.05° cell resolution, although the formatted product also offers monthly global BA summaries at 0.25° resolution. Therefore, the original product is at 0.05°.

The methodology and statistical metrics are consistent and meet the expected standards, but more detail is needed in the development. Where can the input data be obtained? How and where was the detrended (Seasonal and Trend Decomposition) performed?

Minor comments

The introduction does not mention anything about the burned area database and the statistical analysis used in the analysis. A simple explanation could give more contextual information to better understand the importance of the study.

Manuscript, L44-48. The explanation is a long and confusing sentence. The explanation of the CTs has to be extended to better understand what you are studying.

Manuscript, L60. What CT domains? That sentence is confusing.

Manuscript, L273-274. Change Font to: "in the region"

Manuscript, L312. Reference in the proper format.

Supplementary materials, line 69: The reference (1) does not correspond to the R software.

Manuscript, L138, and Supplementary materials, line 213: Same sentence, different references.

Please be consistent. "About 70% of BA": both the manuscript reference and the supplementary materials reference may be used. Include van der Werf et al. 2018 reference in the manuscript, at the end of the sentence.

Reviewer #3 (Remarks to the Author):

Review of "Climate teleconnections modulate global burned area".

The authors propose to determine the main teleconnections patterns impacting interannual variability of burned area during peak fire season, globally. A number of indices, representative of numerous climate and weather events, are used to correlate with burned area. From the correlation results, authors determine the teleconnection patterns most relevant to 7 main areas defined from a cluster analysis.

The study combines many different climate phenomena in one simplified correlation analysis, lacking appropriate discussion and interpretation of results. Correlation alone is not causation and the authors do not offer explanation for behaviors that are contrary to well established understanding of how some remote climate drivers impact fires.

I do not recommend this article for publication and I present more detailed arguments in the following paragraphs.

Comments:

1)Line 52: Although the relationships between wildfires and CTs have been studied in several regions across the globe 3-5,11, and the effects of CTs on weather patterns are expected to become more extreme in the future as a result of climate change4,7, the global CT drivers of BA across the worlds' biomes remain unknown.

Reviewer's Comment: I disagree with this sentence, but perhaps I misunderstood. The authors themselves have listed studies that show how remote climate drivers (such as ENSO) can impact wildfires.

2)Line 100: PNA appeared to have mainly influenced BA synchronously, whereas SAM, ENSO and NAO were found to predominantly have lagged effects.

Reviewer's Comment: The sentence above is in relation to Fig. 2A. PNA is one important mode of variability in the northern hemisphere high-latitudes. How do authors explain significant correlation of PNA with fires in southern equatorial Africa during its peak fire season (JJA- Figure 1B) when PNA is not active? From NOAA:

"The Pacific-North America (PNA) pattern is one of the most prominent modes of low-frequency variability in the Northern Hemisphere extratropics, appearing in all months except June and July". (<https://www.ncdc.noaa.gov/teleconnections/pna/>)

3)In reference to Table 1 and Figure 2.

Reviewer's Comment: How do the authors conciliate combining in one single correlation analysis CT drivers that vary on the scale of days and weeks (ex: SAM, NAO) and those that vary on the scale of several decades (ex:PDO)? Also, the significance of correlation of timeseries that are serially correlated (like PDO) tends to be inflated. How did the authors account for that in their study? I refer the authors to:

Ebisuzaki, Wesley. "A method to estimate the statistical significance of a correlation when the data are serially correlated." *Journal of Climate* 10.9 (1997): 2147-2153.

4)Line 144: Savanna fires responded mostly to ENSO, PDO, SAM and EA in the northern hemisphere (CTD 3).

Reviewer's Comment: Similar to comment #2 above. How do authors account for the Southern Annular Mode (SAM), which is a mode of variability in the southern hemisphere high-latitudes, influence on northern hemisphere fires. Is there any study connecting this cross-hemispheric effect?

5)Line 181: Unexpectedly, our results indicated that there appeared to be decreases in wildfire BA under +ENSO despite documented precipitation deficits in austral summer19. This fact may be explained by the effects of the two different types of +ENSO suggested by Li et al 2011.

Reviewer's Comment: In relation to Figure 3B. The El Niño phenomenon in the Amazon (+ENSO) causes reduced precipitation, thus it should be expected to positively correlate with fires, as mentioned by the authors. But the results show the opposite behavior, prompting the authors to explain it by referring to different types of ENSO. Although different types of ENSO may cause slightly different precipitation patterns in the Amazon, it would not be enough to explain the contrary behavior (negative correlation denoting ENSO+/drought causing a decrease in fires) observed and I consider this argument not appropriate.

Perhaps a better explanation is that the analysis of CT-fire clusters, pools all regions together to derive Figure 3B. The Amazon is pooled together with equatorial Africa, southeast Asia, parts of Siberia and the US. Fires in these regions should all relate very differently to remote climate drivers and the behavior summarized for cluster 2 is not necessarily representative of the Amazon.

Supplementary Material

Line 26: Using a 3-month moving average, we analyzed the main climate teleconnections (CTs) modulating weather patterns at global scale known to influence wildfire activity from regional to continental scales²⁻⁶, namely:

Reviewer's Comment: How were these CT selected? Based on literature? Although there are some new potential weather/climate drivers of fire variability being considered in this work (AO, NAO, SAM), there are some well-known remote drivers of fires in the tropical Atlantic (AMO, NTA) not included in the study (see references below).

Fernandes, Katia, et al. "North Tropical Atlantic influence on western Amazon fire season variability." *Geophysical Research Letters* 38.12 (2011).

Chen, Yang, et al. "Forecasting fire season severity in South America using sea surface temperature anomalies." *Science* 334.6057 (2011): 787-791.

da Silva, Sonaira Souza, et al. "Dynamics of forest fires in the southwestern Amazon." *Forest Ecology and Management* 424 (2018): 312-322.

Line 46: Arctic Oscillation (AO) teleconnection is highly correlated with the NAO and is derived as a function of winter sea level pressure (geopotential heights) and surface air temperature fluctuations over mid to high latitudes (>20°N) in the Northern Hemisphere¹¹

Reviewer's Comment: Why use an index (AO) that is highly correlated to another index (NAO)?

The mechanisms explaining changes in weather conditions in distant regions are obviously different for all CTs despite that underlying associations were found. It is really difficult to isolate the effect of a specific CT on burned area from others due to these underlying correlations. However, note that we only considered the CT with the highest correlation with BA at pixel level to evaluate their contribution to global BA.

We agree with the reviewer that several CTs may influence BA at the same time. That is the reason why we analyzed the associations of CTs influencing BA at pixel level through the clustering analysis presented in the main manuscript (Fig 3) where marked regional partners were identified across the globe.

Figure S1. Relationships among climate teleconnections (CT) from 1982 to 2018 using Principal Component Analysis (PCA). A) First versus second PCA component (34.0% of variance); B) third versus fourth PCA component (20.2% of variance). Color indicates the importance of the component for each variable as the square cosine.

Some of the CTs you mentioned are widely recognized as CT but some are not. Some patterns have both regional impacts and teleconnections. Please specify how do you distinguish the impacts of regional impacts and teleconnections. For example, the enso and pdo you selected are highly correlated.

The nature of CTs allows significant climate remote responses that exist between geographically distant or non-contiguous regions, either concurrent with or time lagged. This is one of the reasons why we developed this research work since it allows us to better understand the fire-CT linkages at regional scales on Earth considering lagged effects and the influence of each CT modulating BA at pixel level (0.5° pixel resolution). Therefore, the key point of our work is to better understand the CT-fire relationships between geographically distant or non-contiguous regions. This was reflected in Figure 2 and 3 of the main manuscript with the significance of fire-CT correlations and their aggregation into CT domains.

Please explain the selection of these ct patterns. For example, why do you selected pdo but not amo? Why do you consider SAM but not NAM? There are also other ct patterns.

Thanks for your comment, also raised by reviewer 3. We selected the CTs based on an extensive review of scientific literature, most of the papers are cited in the Supplementary Materials. There are several CT indexes across the globe not selected in our analysis due to several reasons: low signal of the CT index, lack of relevance in previous studies or low influence of the CT index on weather patterns at global scale. Our current selection is in agreement with recent research studying the linkages between CTs and fire (Shi and Touge, 2022; Chen et al., 2016) and CT and weather (Martens et al., 2018).

We agree with the reviewer in relation to the AMO since previous research identified how this CT modulated burned area in several regions (for instance, Cardil et al., 2019 in Southern California). We did not include it in our preliminary analysis due to its low frequency signal. However, we extended the time series of both CTs and BA in the revised version of the manuscript and, now, our analysis reveals its prominent role explaining BA in many regions on Earth.

Note that we included additional CT indexes in the revised version of the manuscript based on reviewer 3 suggestions and the findings of Shi and Touge (2022). We finally added the following CT indexes: AMO, TNA, TSA, EP and WP.

Shi, K., Touge, Y. Characterization of global wildfire burned area spatiotemporal patterns and underlying climatic causes. *Sci Rep* 12, 644 (2022). <https://doi.org/10.1038/s41598-021-04726-2>

Chen, Y., Morton, D. C., Andela, N., Giglio, L. & Randerson, J. T. How much global burned area can be forecast on seasonal time scales using sea surface temperatures? *Environ. Res. Lett.* 11, 045001 (2016).

Martens, B., Waegeman, W., Dorigo, W.A. et al. Terrestrial evaporation response to modes of climate variability. *npj Clim Atmos Sci* 1, 43 (2018). <https://doi.org/10.1038/s41612-018-0053-5>

Line 294, they are indeed negatively correlated. How to disentangle their impacts?

We have included more analysis in the supplementary materials assessing how CTs are correlated. ENSO/SAM were not correlated during the study period ($P < 0.05$). Despite this correlation the mechanisms explaining changes in weather conditions in distant regions are obviously different between SAM and ENSO and, therefore, it is key to study them independently. We recognize it is really difficult to isolate the effect of a specific CT on burned area due to these underlying slight correlations.

Dätwyler, C., Grosjean, M., Steiger, N. J. & Neukom, R. Teleconnections and relationship between the El Niño–Southern Oscillation (ENSO) and the Southern Annular Mode (SAM) in reconstructions and models over the past millennium. *Clim. Past* 16, 743–756 (2020).

Line 303-304, it is difficult to understand that the human induced ignition can amplify the relationships between ct and climate. The human caused ignitions can increase the BA but not their linkages with climate.

We agree with the reviewer and we decided to remove this sentence from the manuscript.

Please add more details in the figure captions.

Done. We have improved the figure captions.

It is suggested to add more details on the methods and data. Different fire datasets have different quality of the BA.

Reviewer 2 also requested to add more details on the fire data we used. This was addressed in the section “S.1.1.1 Fire data” in the supplementary materials.

Also, we improved the description of the methods used in the statistical analysis. Please, see the supplementary materials (S.1.1.3. Statistical analysis).

Reviewer #2 (Remarks to the Author):

Summary

The manuscript presents a global relationship between weather conditions and fire activity, which vary across continents and biomes. The study find that the 65% of global burned area (peak of the fire season) is modulated by seven global climate connections. The global analysis focuses on regional patterns, appropriately using the burned area database. The method is based on consistent statistical metrics, offering results $p < 0.05$ which is statistically significant. In addition, the study has been widely discussed with an appropriate literature. Therefore, the manuscript presents a novel and relevant study/results that are very interesting for the science community. It helps us improve our understanding of climate and its relationship to fires.

Major comments

The main comment is about the burned area dataset used in the analysis. In the study, FireCCILT10 is mentioned as the product used, but the bibliographic reference corresponds to the FireCCILT11 product. Please, what database has been used? FireCCILT10 is a deprecated Beta version with serious limitations that should no longer be used. On the contrary, FireCCILT11 is a consistent version that improves and fixes the problems of the previous Beta version. The FireCCILT11 product is freely available and suitable for this study. If FireCCILT11 is used, as expected, please changed the information in the manuscript and in the Supplementary Materials.

Thanks for your comment. You are right, we used the FireCCILT11 product. We corrected the product name in both the manuscript and Supplementary Materials.

As a suggestion, FireCCILT11 covers the period from 1982 to 2018 (excluding 1994). Why did the study start in 1995? and not in 1982? More years should be interesting for this manuscript and could offer strong patterns. Despite that, the current time series (1995-2018) shows very interesting results.

We followed your guidance and extended the study period from 1982 to 2018. This allowed us to include low-frequency CT modes such as AMO in the new version of the

manuscript. The findings are more significant and consistent in the new version of the manuscript.

Regarding the product nomenclature, the correct way to reference the product, recommended by Fire_cci project (European Space Agency), is FireCCILT11 and not FireCCILT11v1.1. The number 11 corresponds to the version number.

Ok, thank you. Addressed. We use “FireCCILT11” in the new version of the manuscript.

In Supplementary materials, line 17-18: The current website of the Fire_cci (ESA) project is: <https://climate.esa.int/en/projects/fire/>. Please change it.

Done.

In Supplementary materials, line 19-20: It is necessary to clarify that AVHRR is a sensor and LTDR is the dataset. Therefore, the LTDR dataset is based on images of the AVHRR sensors. **Addressed as follows: “The product was assembled using spectral information from the AVHRR sensor (Advanced Very High Resolution Radiometer) and Land Long Term Data Record (LTDR) version 5 dataset produced by NASA”.**

In Supplementary materials, line 21-22: Clarify that FireCCILT11 includes monthly global BA data estimates at 0.05° cell resolution, although the formatted product also offers monthly global BA summaries at 0.25° resolution. Therefore, the original product is at 0.05°.

We agree with you. We have modified the sentence as follows: “The FireCCILT11 Fire_cci v1.0 includes monthly global BA data estimates at 0.05° cell resolution. Data was resampled at 0.5° resolution to facilitate data handling and processing while enhancing pattern recognition.”

The methodology and statistical metrics are consistent and meet the expected standards, but more detail is needed in the development. Where can the input data be obtained? How and where was the detrended (Seasonal and Trend Decomposition) performed?

Addressed in the supplementary materials. The Seasonal Trend Decomposition (STL) procedure was applied to both CT and burned area monthly time series. Time series were submitted to STL before computing the lagged averages, splitting the series into its ‘trend’, ‘seasonal’ and ‘random’ components, retaining only the latter two, i.e., removing the tendency in the time series. The procedure was conducted at the pixel level in the case of burned area data, since it was the only information that varies spatially. Once the STL was performed, the detrended time series were aggregated as a 3-month average and correlation is calculated using the 3-month period of burned area during the fire season peak and the corresponding 3-month lagged interval (0, 3, 6 and 9 months before the fire season peak) of CT indexes.

Minor comments

The introduction does not mention anything about the burned area database and the statistical analysis used in the analysis. A simple explanation could give more contextual information to better understand the importance of the study.

We agree. We have now added the reference to the burned area database and mentioned the statistical analysis used in our analysis as suggested.

“We explore the complex associations through cross-correlation analysis between the major modes of SSTs and CTs (Fig.1 and S.1.1.2) driving large-scale atmospheric anomalies and global BA from 1982 to 2018 at 0.5° pixel resolution (Fig. 1; FireCCILT11 global BA product13), translating them into transboundary CT domains (CTD).”

Manuscript, L44-48. The explanation is a long and confusing sentence. The explanation of the CTs has to be extended to better understand what you are studying.

We clarified and extended the definitions of CTs.

Manuscript, L60. What CT domains? That sentence is confusing.

Addressed.

Manuscript, L273-274. Change Font to: “in the region”

Addressed.

Manuscript, L312. Reference in the proper format.

Addressed.

Supplementary materials, line 69: The reference (1) does not correspond to the R software.

Addressed.

Manuscript, L138, and Supplementary materials, line 213: Same sentence, different references. Please be consistent. “About 70% of BA”: both the manuscript reference and the supplementary materials reference may be used. Include van der Werf et al. 2018 reference in the manuscript, at the end of the sentence.

Done, addressed.

Reviewer #3 (Remarks to the Author):

Review of “Climate teleconnections modulate global burned area”.

The authors propose to determine the main teleconnections patterns impacting interannual variability of burned area during peak fire season, globally. A number of indices, representative of numerous climate and weather events, are used to correlate with burned area. From the correlation results, authors determine the teleconnection patterns most relevant to 7 main areas defined from a cluster analysis.

The study combines many different climate phenomena in one simplified correlation analysis, lacking appropriate discussion and interpretation of results. Correlation alone is not causation and the authors do not offer explanation for behaviors that are contrary to well established understanding of how some remote climate drivers impact fires.

Thank you very much for your comment. We have polished the manuscript to avoid causation and explain all CT-fire relationships found in our analysis.

Our study analyzed the correlations between BA and the different CT indexes independently to identify CT-fire patterns at global scale. Also, we clustered the CT-fire into CT domains to better interpret the associations between CT indexes leading to increased/decreased BA on interannual timescales.

We have made an effort to further explain all these CT-fire relationships and further justify our findings. The percentage of BA modulated by CTs is high, given that fire is not just controlled by global climate but also regionally by land management, fire suppression capabilities and practices. Obviously, it is challenging to consider local and regional factors at global scale but we have done our best to interpret the CT-fire patterns by continent.

I do not recommend this article for publication and I present more detailed arguments in the following paragraphs.

Comments:

1)Line 52: Although the relationships between wildfires and CTs have been studied in several regions across the globe 3–5,11, and the effects of CTs on weather patterns are expected to become more extreme in the future as a result of climate change^{4,7}, the global CT drivers of BA across the worlds' biomes remain unknown.

Reviewer's Comment: I disagree with this sentence, but perhaps I misunderstood. The authors themselves have listed studies that show how remote climate drivers (such as ENSO) can impact wildfires.

We rephrased the text. Indeed, a number of studies have already been performed on regional to global scales but no synthesis has been introduced so far. By synthesizing the influence of a range of CTs on BA including synchronous and lagged effects in the different biomes by continent, we hope this study will provide valuable insights to the scientific community.

2)Line 100: PNA appeared to have mainly influenced BA synchronously, whereas SAM, ENSO and NAO were found to predominantly have lagged effects.

Reviewer's Comment: The sentence above is in relation to Fig. 2A. PNA is one important mode of variability in the northern hemisphere high-latitudes. How do authors explain significant correlation of PNA with fires in southern equatorial Africa during its peak fire season (JJA- Figure 1B) when PNA is not active? From NOAA:

“The Pacific-North America (PNA) pattern is one of the most prominent modes of low-frequency variability in the Northern Hemisphere extratropics, appearing in all months except June and July”. (<https://www.ncdc.noaa.gov/teleconnections/pna/>)

We correlated the PNA with several weather variables in the study period and we found strong PNA-precipitation associations (1982-2018). The most relevant PNA-fire association now is at 3-month lag from March to May, being the fire season from June to

August. +PNA (lag 3) was associated with decreasing BA as it is potentially linked to above average convective precipitation over central Africa during the monsoon season.

3)In reference to Table 1 and Figure 2.

Reviewer's Comment: How do the authors conciliate combining in one single correlation analysis CT drivers that vary on the scale of days and weeks (ex: SAM, NAO) and those that vary on the scale of several decades (ex:PDO)? Also, the significance of correlation of timeseries that are serially correlated (like PDO) tends to be inflated. How did the authors account for that in their study? I refer the authors to:

To overcome this potential limitation data were aggregated/averaged at the seasonal scale and correlations were computed at the interannual time scale. Furthermore the STL should help in homogenizing the time series in terms of differences in frequency of the signal, but the fact that we focus on interannual time scales enables the comparison. Nonetheless, this effect (serial autocorrelation) is still partially seen in some CT patterns, which we acknowledge in the discussion. For instance, the new correlations calculated for the AMO -the slowest reacting CT- showed slim differences per lag, thus indicating the persistence of serial correlation. But in the case of the TNA the effect was considerably less prominent.

4)Line 144: Savanna fires responded mostly to ENSO, PDO, SAM and EA in the northern hemisphere (CTD 3).

Reviewer's Comment: Similar to comment #2 above. How do authors account for the Southern Annular Mode (SAM), which is a mode of variability in the southern hemisphere high-latitudes, influence on northern hemisphere fires. Is there any study connecting this cross-hemispheric effect?

Reviewer 3 raised an interesting question regarding the modulation of SAM on BA in the northern hemisphere. As expected by us and the reviewer, we found that SAM mostly influences BA in the southern hemisphere, especially in the Savannas of Cerrado (South America) and Africa and several regions of Australia.

However, we decided to run the statistical analysis globally for all CT indexes to further explore the CT-fire relationships in both hemispheres. For instance, SAM may also influence specific areas of the northern hemisphere. Liu et al., 2015 highlighted how "The SAM contributes not only to climate change in the SH but also to climate anomalies in the Northern Hemisphere (Zheng et al. 2014; Zheng et al. 2015; Li 2016). In recent years, the influence of the SAM on NH climate anomalies has received considerable attention."

Liu, T., Li, J., & Zheng, F. (2015). Influence of the Boreal Autumn Southern Annular Mode on Winter Precipitation over Land in the Northern Hemisphere, Journal of Climate, 28(22), 8825-8839. Retrieved Aug 12, 2022, from <https://journals.ametsoc.org/view/journals/clim/28/22/jcli-d-14-00704.1.xml>

Zheng, F., J. P. Li, L. Wang, F. Xie, and X. Li, 2015: Cross-seasonal influence of the December–February Southern Hemisphere annular mode on March–May meridional circulation and precipitation. *J. Climate*, 28, 6859–6881, doi:10.1175/JCLI-D-14-00515.1.

Zheng, F., J. P. Li, and T. Liu, 2014: Some advances in studies of the climatic impacts of the Southern Hemisphere annular mode. *J. Meteor. Res.*, 28, 820–835, doi:10.1007/s13351-014-4079-2.

5)Line 181: Unexpectedly, our results indicated that there appeared to be decreases in wildfire BA under +ENSO despite documented precipitation deficits in austral summer19. This fact may be explained by the effects of the two different types of +ENSO suggested by Li et al 2011.

Reviewer's Comment: In relation to Figure 3B. The El Niño phenomenon in the Amazon (+ENSO) causes reduced precipitation, thus it should be expected to positively correlate with fires, as mentioned by the authors. But the results show the opposite behavior, prompting the authors to explain it by referring to different types of ENSO. Although different types of ENSO may cause slightly different precipitation patterns in the Amazon, it would not be enough to explain the contrary behavior (negative correlation denoting ENSO+/drought causing a decrease in fires) observed and I consider this argument not appropriate.

Perhaps a better explanation is that the analysis of CT-fire clusters, pools all regions together to derive Figure 3B. The Amazon is pooled together with equatorial Africa, southeast Asia, parts of Siberia and the US. Fires in these regions should all relate very differently to remote climate drivers and the behavior summarized for cluster 2 is not necessarily representative of the Amazon.

Thanks for your suggestion. The new results with the extended time series (1982-2018) show no significant correlation between ENSO and BA in the Amazon. Therefore, we have removed this sentence from the main manuscript. However, we further investigated the reviewers' concern and we concluded that:

- **This was not an issue related to our methodology as the reviewer suggested since we calculated the correlation between ENSO and BA independently to other CT modes. This process was not related to the hierarchical clustering approach to derive Fig 3 that includes the Amazon together with equatorial Africa and southeast Asia.**
- **After considering this comment, we agree with the reviewer that different types of ENSO may not explain the negative correlation between ENSO and BA we found in the original version of the manuscript (1995-2018).**
- **There is discussion about how ENSO influences weather and BA in the Amazon. In fact, in recent years, The years considered as La Niña were those with the highest BA due to the interaction with other variables (See Ferreira Barbosa et al., 2021 for more details.**

Ferreira Barbosa, M. L., Delgado, R. C., Forsad de Andrade, C., Teodoro, P. E., Silva Junior, C. A., Wanderley, H. S., & Capristo-Silva, G. F. (2021). Recent trends in the fire dynamics in Brazilian Legal Amazon: Interaction between the ENSO phenomenon, climate and land

use. *Environmental Development*, 39, 100648.
<https://doi.org/https://doi.org/10.1016/j.envdev.2021.100648>

- Also, Martens et al., (2018) found that higher rates of evapotranspiration may occur in the Amazon during La Niña. All this discussion has been reflected in the supplementary materials (Americas section).

Martens, B., Waegeman, W., Dorigo, W.A. et al. Terrestrial evaporation response to modes of climate variability. *npj Clim Atmos Sci* 1, 43 (2018). <https://doi.org/10.1038/s41612-018-0053-5>

Supplementary Material

Line 26: Using a 3-month moving average, we analyzed the main climate teleconnections (CTs) modulating weather patterns at global scale known to influence wildfire activity from regional to continental scales^{2–6}, namely:

Reviewer's Comment: How were these CT selected? Based on literature? Although there are some new potential weather/climate drivers of fire variability being considered in this work (AO, NAO, SAM), there are some well-known remote drivers of fires in the tropical Atlantic (AMO, NTA) not included in the study (see references below).

Fernandes, Katia, et al. "North Tropical Atlantic influence on western Amazon fire season variability." *Geophysical Research Letters* 38.12 (2011).

Chen, Yang, et al. "Forecasting fire season severity in South America using sea surface temperature anomalies." *Science* 334.6057 (2011): 787-791.

da Silva, Sonaira Souza, et al. "Dynamics of forest fires in the southwestern Amazon." *Forest Ecology and Management* 424 (2018): 312-322.

We selected the CTs based on an extensive review of scientific literature, most of the papers are cited in the Supplementary Materials. There are several CT indexes across the globe not selected in our analysis due to several reasons: low signal of the CT index, lack of relevance in previous studies or low influence of the CT index on weather patterns at global scale. Our current selection is in agreement with recent research studying the linkages between CTs and fire (Shi and Touge, 2022; Chen et al., 2016) and CT and weather (Martens et al., 2018).

We agree with the reviewer in relation to the AMO since previous research identified how this CT modulated burned area in several regions (for instance, Cardil et al., 2019 in Southern California). We did not include it in our preliminary analysis due to its low frequency signal. However, we extended the time series of both CTs and BA in the revised version of the manuscript and, now, our analysis reveals its prominent role explaining BA in many regions on Earth.

Note that we included additional CT indexes in the revised version of the manuscript based on reviewer 3 suggestions and the findings of Shi and Touge (2022). We finally added the following indexes: AMO, TNA, TSA, EP and WP.

Shi, K., Touge, Y. Characterization of global wildfire burned area spatiotemporal patterns and underlying climatic causes. *Sci Rep* 12, 644 (2022). <https://doi.org/10.1038/s41598-021-04726-2>

Chen, Y., Morton, D. C., Andela, N., Giglio, L. & Randerson, J. T. How much global burned area can be forecast on seasonal time scales using sea surface temperatures? *Environ. Res. Lett.* 11, 045001 (2016).

Martens, B., Waegeman, W., Dorigo, W.A. et al. Terrestrial evaporation response to modes of climate variability. *npj Clim Atmos Sci* 1, 43 (2018). <https://doi.org/10.1038/s41612-018-0053-5>

Line 46: Arctic Oscillation (AO) teleconnection is highly correlated with the NAO and is derived as a function of winter sea level pressure (geopotential heights) and surface air temperature fluctuations over mid to high latitudes (>20°N) in the Northern Hemisphere¹¹

Reviewer's Comment: Why use an index (AO) that is highly correlated to another index (NAO)?

We have included a PCA analysis to show how the CT indexes are correlated. AO and NAO are correlated based on this analysis. Also, previous research highlighted this strong relationship between both indexes. However, there is no consensus on the links between them and if the effects on weather may vary spatially (Baez et al., 2005). Both the NAO and the AO may be either reflections in the troposphere from the same common cause or different phenomena with independent and complementary effects.

For instance, “Frias et al. [24] analyzed the impact of the NAO and AO on the Iberian water resources, and they observed that NAO could explain the inter-annual variability of southern Iberia, while AO was better associated with river flow in the northern basins. This suggests that NAO and AO may have different effects on the cycling of the freshwater runoff, which might have a delayed effect on SST in the eastern Mediterranean Sea.”. Obviously, this could also influence other weather parameters and, subsequently, burned area.

Finally, we decided to keep both CTs in our analysis since our objective is to disentangle the effects of each CT mode modulating BA and not creating a parsimonious model to predict BA.

Frias T, Trigo R, Valente M, Pires C (2005) The impact of the NAO and AO on the Iberian water resources. *Geophys Res Abs* 7: 1607–7962/gra/EGU05-A-04278.

Báez JC, Gimeno L, Gómez-Gesteira M, Ferri-Yáñez F, Real R. Combined effects of the North Atlantic Oscillation and the Arctic Oscillation on sea surface temperature in the Alborán Sea. *PLoS One*. 2013 Apr 18;8(4):e62201. doi: 10.1371/journal.pone.0062201. PMID: 23638005; PMCID: PMC3630154.

Reviewer comments, second round -

Reviewer #1 (Remarks to the Author):

I have checked for the revision and satisfied with their efforts. I agree with publication in its current form.

Reviewer #2 (Remarks to the Author):

The authors made an effort to revise the manuscript, and my comments have been satisfactorily addressed. The paper is clear and well referenced, showing very interesting results. They used the burned area database appropriately. The method is based on consistent statistical metrics and is described with detail. The results are well explained and are supported by clear figures. Furthermore, the study has been widely discussed with an appropriate literature. Therefore, the manuscript presents a novel and relevant study/results that are of great interest to the scientific community. It helps us to improve our understanding of climate and its relationship with fires. In my opinion, I recommend this article for publication.

RESPONSE TO REFEREE LETTERS

Reviewer #1 (Remarks to the Author):

I have checked for the revision and satisfied with their efforts. I agree with publication in its current form.

Reviewer #2 (Remarks to the Author):

The authors made an effort to revise the manuscript, and my comments have been satisfactorily addressed. The paper is clear and well referenced, showing very interesting results. They used the burned area database appropriately. The method is based on consistent statistical metrics and is described with detail. The results are well explained and are supported by clear figures. Furthermore, the study has been widely discussed with an appropriate literature. Therefore, the manuscript presents a novel and relevant study/results that are of great interest to the scientific community. It helps us to improve our understanding of climate and its relationship with fires. In my opinion, I recommend this article for publication.

Authors: We really appreciate the comments and suggestions raised by all reviewers and editor during the peer review process. The comments were very constructive and useful to improve our manuscript.